

# A niche for diverse cable bacteria in continental margin sediments overlain by oxygen-deficient waters

Caroline P. Slomp[1,2], Martijn Hermans[2,3], Niels A.G.M. van Helmond[1,2], Silke Severmann[4], James McManus[5], Marit R. van Erk[1], Sairah Malkin[6]

[1]Department of Microbiology, Radboud University, Nijmegen, the Netherlands
[2]Department of Earth Sciences, Utrecht University, Utrecht, the Netherlands
5   [3]Baltic Sea Centre, Stockholm University, Stockholm, Sweden
[4]Department of Marine and Coastal Science, Rutgers University, New Brunswick, NJ, United States
[5]Bigelow Laboratory for Ocean Sciences, East Boothbay, ME, United States
[6]Horn Point Laboratory, University of Maryland Center for Environmental Science (UMCES), Cambridge, United States

10   *Correspondence to*: Caroline P. Slomp (caroline.slomp@ru.nl)

**Abstract.** Since the discovery of cable bacteria more than a decade ago, these multi-cellular, filamentous sulfur-oxidizing bacteria have been found in a range of sedimentary environments. However, their abundance, diversity and activity in continental margin sediments overlain by oxygen-deficient waters at water depths >100 m remain poorly known. Here we address this by studying five basins along the coasts of California and Mexico. All sediments are organic carbon rich (2.5 – 15   7.5 wt%) and characterized by active iron and sulfur cycling. Nitrate is present in the bottom water at all sites. Results of fluorescence in-situ hybridization (FISH) indicate a low areal abundance of cable bacteria (0.2 to 19 m cm$^{-2}$) in sediments of the hypoxic San Clemente, Catalina and San Pedro basins and anoxic San Blas basin. In the anoxic Soledad basin, in contrast, we found abundant cable bacteria near the sediment surface (129 m cm$^{-2}$). DNA amplicon sequencing detected cable bacteria reads in sediments of the hypoxic San Pedro, and anoxic Soledad and San Blas basins. Phylogenetic analysis indicated that the 20   diversity of the amplicon sequence variants (ASVs) was spread across the *Candidatus* Electrothrix lineage, including multiple ASVs closely related to *Electrothrix gigas*, a recently discovered species of giant cable bacteria. Additionally, multiple sequences retrieved from the Soledad and San Blas basins revealed affiliation with a clade sister to *Electrothrix*, hypothesized as a novel genus, suggesting possible relic or novel adaptations of cable bacteria to these anoxic and nitrogenous environments. The areal abundance of cable bacteria was negatively related to sediment Fe/S ratios suggesting a control by sulfide availability. 25   Free sulfide in the porewater was only detected at the anoxic Soledad site, however. Micro-profiling of pH and electric potential point towards a lack of cable bacteria activity at the time of sampling, possibly due to a limitation by a suitable electron donor and/or acceptor. Periodically enhanced organic matter input and associated sulfate reduction and/or inflows of oxic water could alleviate the deficiency, creating the observed niche for diverse cable bacteria.



# 1 Introduction

Cable bacteria are multicellular, filamentous bacteria that can couple the oxidation of sulfide to the reduction of oxygen ($O_2$), nitrate ($NO_3^-$) or nitrite ($NO_2^-$) by transporting electrons over centimetre-scale distances (Nielsen et al., 2010; Pfeffer et al., 2012; Marzocchi et al., 2014). When metabolically active, cable bacteria strongly affect porewater pH thereby altering the sedimentary cycling of sulfur, iron, manganese, calcium and phosphorus (Risgaard-Petersen et al., 2012; Nielsen and Risgaard-

Petersen, 2015; Seitaj et al., 2015; Sulu-Gambari et al., 2016). Since cable bacteria can affect the release of hydrogen sulfide ($H_2S$) and the nutrient phosphate from sediments to overlying waters, this may impact ecosystem functioning (Seitaj et al., 2015; Sulu-Gambari et al., 2016). Cable bacteria have been observed in a wide range of brackish and marine environments (Burdorf et al., 2017; Dam et al., 2021), including sulfidic surface sediments in the coastal zone of Peru and Chile where DNA data point to their presence (Fonseca et al., 2022). Little is known about their diversity, abundance and activity in sediments

of oxygen-deficient waters along other continental margins.

Cable bacteria belong to the family *Desulfobulbaceae.* Two genera, *Candidatus* Electronema (primarily freshwater) and *Candidatus* Electrothrix (primarily brackish and marine), are formally described (Trojan et al., 2016; Dam et al., 2021; Plum-Jensen et al., 2024). Three additional genera (AR3, AR4, SI2) have recently been hypothesized based on genomic analysis of individual filaments or sequence similarity of near full length 16S rRNA genes, but are not yet described (Geelhoed

et al., 2023, Ley et al., 2024). The ecological niche of cable bacteria is thought to be primarily determined by the availability of $H_2S$ or iron monosulfide (FeS) as an electron donor and oxygen or nitrate as an electron acceptor (Risgaard-Petersen et al., 2012; Burdorf et al., 2017). Cable bacteria activity is typically reflected in the development of a suboxic zone (i.e. a zone devoid of both $O_2$ and $H_2S$), a pH maximum near the sediment-water interface and a pH minimum in the suboxic zone (Pfeffer et al., 2012; Meysman et al., 2015) and an increase in the electric potential with sediment depth (Damgaard et al., 2014).

Sediments at a given location frequently harbor multiple cable bacteria species but the factors contributing to this diversity are not yet well understood (Dam et al., 2021; Fonseca et al., 2022; Liau et al., 2022; Geelhoed et al., 2023;). Cable bacteria interact with other members of the microbial community, and depending on sediment depth and temporal environmental changes, may compete with and/or promote the activity of other microbes (Vasquez-Cardenas et al., 2015; Liu et al., 2021; Liau et al., 2022, Bjerg et al., 2023). In seasonally hypoxic coastal systems ($[O_2] < 63$ µmol $L^{-1}$), cable bacteria

have been shown to compete for a similar niche as sulfur-oxidizing *Beggiatoaceae* (Seitaj et al., 2015; Sulu-Gambari et al., 2016; Hermans et al., 2019; Malkin et al., 2022).

Sulfur-oxidizing bacteria are a well-known feature in continental margin sediments in Oxygen Deficient Zones (ODZs) in the Atlantic and Pacific Oceans (Chong et al., 2012; Callbeck et al., 2021). In these regions, coastal upwelling drives a high primary productivity and a high flux of organic matter to the sediment, thereby promoting high rates of sulfate reduction

near the sediment-water interface. Nitrate is typically present in the bottom water, while oxygen concentrations vary with water depth but also temporally, due to periodic (often seasonal) variations in organic matter and oxygen supply (e.g. Peng et al., 2024; Yousavich et al., 2024). Hence, bottom waters in ODZs typically range from hypoxic to anoxic (i.e. $[O_2] = 0$ µmol $L^{-1}$).



*Thioploca* and *Beggiatoaceae* are common genera of sulfur-oxidizing bacteria in sediments in ODZs along the coasts of South and North America (Jørgensen, 2021). Along the Namibian coastline, in contrast, the sulfur-oxidizer *Thiomargarita*

*namibiensis* generally dominates, possibly because it can tolerate frequent sediment resuspension, which is a feature of the Namibian shelf (Schulz et al., 1999). *Beggiatoaceae*, *Thioploca* and *Thiomargarita* are generally referred to as "giant" sulfur bacteria because of their large cell sizes (Schulz and Jørgensen, 2001). They are at a competitive advantage over many other sulfur-oxidizers because they can store nitrate internally in vacuoles and use that to oxidize hydrogen sulfide generated at greater depths in the sediment, frequently by motility. The presence of *Ca.* Electrothrix in sediments underlying the Peruvian

and Chilian ODZ (Fonseca et al., 2022) suggest that cable bacteria might occupy a similar niche, with oxygen-poor, nitrate-rich waters.

In this study, we investigated sediments at five sites in the ODZ along the coasts of California and Mexico to assess the abundance, diversity and activity of cable bacteria. The sites capture a range of bottom water $O_2$ concentrations (from hypoxic to anoxic) and rates of organic carbon input and burial (McManus et al., 2006). Detailed geochemical analyses of the

bottom water, porewater and sediment were performed to assess potential electron acceptors and donors for cable bacteria activity. We find evidence for cable bacteria at three of the sites including multiple amplicon sequence variants (ASVs) affiliated with both *Ca.* Electrothrix and a closely related clade. We discuss the potential factors contributing to their occurrence and activity and suggest that further examination of cable bacteria from these hypoxic and anoxic basins could offer insight into their evolutionary history.

**2 Materials and methods**

**2.1 Study sites**

We sampled five sites, each in a different coastal basin, capturing a range of water depths within the upwelling zone along the continental margins of California and Mexico (Fig. 1; Table 1). In this part of the eastern Pacific, the ODZ is present between depths of ~500 and 1000 m (McManus et al., 2006; Chen et al., 2021). Bottom water oxygen concentrations vary from basin

to basin, depending on rates of oxygen consumption in the water column and sediment (Stott et al., 2000) and on topography and sill depth, which determine horizontal water exchange (Berelson et al., 1987; Bruggmann et al., 2023: Bruggmann et al., 2024 and references therein). The residence times of the water in the individual basins are not well-known but bottom water replacement is thought to occur episodically following inflows of denser water over the sills (Berelson et al., 1987; Berelson, 1991). Such inflows may lead to periodic variations in bottom water oxygen concentrations (Bernhard and Buck, 2004; Chong

et al., 2012). The bottom waters in the Californian basins (San Clemente, Catalina, San Pedro) are generally hypoxic, however, while those of the Mexican basins (Soledad and San Blas) are typically anoxic (Table 1; McManus et al., 2006; Bruggmann et al., 2023). Depths of oxygen penetration in the sediment at the hypoxic sites compiled from multiple studies range from 0 to 1.3 cm (Table 1; Bruggmann et al., 2023). Rates of organic carbon ($C_{org}$) burial range from 0.9 to 2.6 mmol m$^{-2}$ d$^{-1}$ and 3.2 to 8.4 mmol m$^{-2}$ d$^{-1}$ at the hypoxic and anoxic sites, respectively (Table 1; McManus et al., 2006).


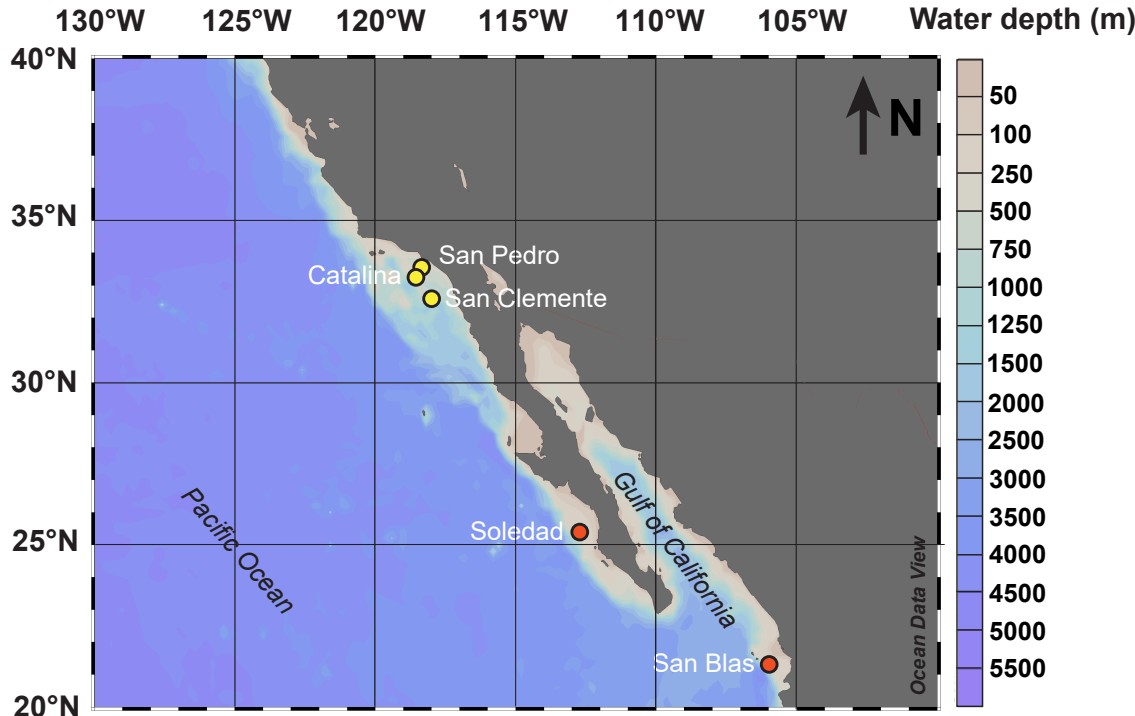

**Figure 1. Sampling locations along the Californian and Mexican margin. This figure was made with Ocean Data View (Schlitzer, 2017). Yellow dots: sites with hypoxic bottom water. Red dots: sites with anoxic bottom water.**


      Of the five silled basins sampled here, the Soledad basin is one of the two shallowest (Table 1) and stands out as the area with the highest burial rate of organic matter (Table 1; McManus et al., 2006). The input of organic matter to the basin varies both seasonally and interannually (Silverberg et al., 2004). Soledad basin is also the only basin of the five where the large sulfur bacterium *Thioploca* has been detected (Chong et al., 2012). *Thioploca* can oxidize hydrogen sulfide with nitrate

and vertically migrates into the sediment, even down to depths of ~6 cm, thereby transporting nitrate in its vacuoles from the bottom water to hydrogen sulfide-bearing sediment layers (Fossing et al., 1995). While there is evidence for the presence of meiofauna (e.g. foraminifera, nematodes; Bernhard and Buck, 2004), no macrofauna have been observed in the sediments of Soledad basin (van Geen et al., 2003; Chong et al., 2012).

**2.2 Water column profiling and sediment and porewater collection**

Samples were obtained during a research expedition with R/V *Oceanus* in May/June of 2018. Water column depth profiles of temperature and salinity were determined with a CTD-system. Dissolved oxygen was measured with a Sea-Bird O₂ sensor



attached to the CTD. Sediment cores with an outer diameter of 10 cm were collected with a multi-corer in a single cast. Bottom water samples were obtained from the overlying water directly after core retrieval. At each site, one core was reserved for
high-resolution microelectrode depth profiling of porewater pH, oxygen, $H_2S$ and electric potential (EP). A second core was sectioned in a glove bag under a nitrogen atmosphere at in-situ temperature to collect sediment and porewater at 0.5 cm depth resolution for the first 10 cm, at 2 cm resolution for the next 10 cm and at 3 to 4 cm resolution for the remainder of the core. A third core was used for enumeration of cable bacteria and 16S rRNA sequencing.

Table 1. Site characteristics. Lat.: Latitude, Long.: Longitude, BW: bottom water, temp: temperature from the CTD, microel.: microelectrode; MAR: mass accumulation rate.

| Site | Lat. (°N) | Long. (°W) | water depth (mbss) | BW temp (°C) | BW O₂* (µmol L⁻¹) CTD | BW O₂ (µmol L⁻¹) CTD; microel. | BW NOₓ (µmol L⁻¹) . | O₂ penetration depth** (cm) | MAR* (mg cm⁻² y⁻¹) | C_org burial* (mmol m⁻² d⁻¹) |
|---|---|---|---|---|---|---|---|---|---|---|
| San Clemente | 32.6 | 118.1 | 2053 | 2.6 | 52 | 50; 50 | 45.2 | 0.63 to 1.28 | 15 | 0.9 |
| Catalina | 33.3 | 118.6 | 1300 | 4.1 | 19 | 12; n.a. | 44.8 | 0 to 0.61 | 14 | 1.2 |
| San Pedro | 33.5 | 118.4 | 896 | 5.2 | 3 to 8 | 2.5; n.a. | 40.1 | 0 to 0.16 | 29 | 2.6 |
| Soledad | 25.2 | 112.7 | 542 | 10.1 | 0 | 2.4; 0 | 28.0 | 0 | 50 | 8.4 |
| San Blas | 21.3 | 106 | 430 | 10.1 | 0 | 0.3; 0 | 23.5 | 0 | 21 | 3.2 |

*: from McManus et al. (2006). **: from Bruggmann et al. (2023). n.a.: not available.

Mats of *Thioploca* or other sulfur-oxidizing bacteria were not observed during the sediment sampling. There was also
no evidence for macrofauna, other than the red crab *Pleuroncodes planipes* observed at the sediment-water interface in several multicores sampled in Soledad Basin (Figure S1). This crab performs diel vertical migration in the water column and can tolerate exceedingly low oxygen concentrations (e.g. Seibel et al., 2018).

High-resolution depth profiles of $O_2$, pH and $H_2S$ were obtained directly upon core retrieval using microelectrodes (50 or 100 µm tip diameter) controlled by a motorized micromanipulator (Unisense A/S, Denmark) as described by Sulu-
Gambari et al. (2016). No $O_2$ profiles for the Catalina and San Pedro basins are available because of a technical problem. Because micro-electrode measurements of $O_2$ in the bottom water for the Soledad and San Blas basins revealed anoxia no further $O_2$ profiling was performed. For depth profiling of EP (500 µm resolution), custom-built electrodes were used as described in Damgaard et al. (2014) and Hermans et al. (2020). The $O_2$ profile for San Clemente was measured immediately after core retrieval, with the measurement being completed within 4 min. Subsequently, pH and $H_2S$ were simultaneously
measured in triplicate for all cores, with each measurement taking 10-15 minutes. Finally, the EP profiles were measured in triplicate for all cores, with each measurement taking 3 minutes. pH values are reported on the total scale. Total $H_2S$ ($\sum H_2S = H_2S + HS^- + S^{2-}$) was calculated as a function of the recorded $H_2S$ and pH values, as well as temperature and salinity (Millero et al., 1988; Jeroschewski et al., 1996).



Sediment slices from the second core were split into two samples: one for porosity determination and one for

porewater collection. Samples for porosity were stored at 4ºC in pre-weighed glass vials. Porewater was collected through centrifugation in 50 mL centrifuge tubes (15 min at 4500 rpm) followed by filtration of the supernatant using 0.45 µm nylon filters under nitrogen. Bottom water and porewater samples were split into subsamples directly after centrifugation under nitrogen for analysis of sulfide ($H_2S$), $NO_x$ (the sum of nitrate and nitrite), ammonium ($NH_4^+$) dissolved manganese (Mn), sulfate ($SO_4^{2-}$), and alkalinity. Bottom water was additionally analyzed for dissolved iron (Fe). Dissolved Fe is assumed to

represent $Fe^{2+}$, while dissolved Mn represents the sum of dissolved $Mn^{2+}$ and $Mn^{3+}$ (Klomp et al. 2025). Samples for $H_2S$ analyses (0.5 mL) were transferred into glass vials containing 2 mL of 2% Zn acetate and stored at 4ºC. Samples for $NO_x$ and $NH_4^+$ were stored at -20ºC, while those for dissolved Mn were acidified with 35% suprapur HCl (10 µl per mL of sample) and stored at 4ºC. Samples for $SO_4^{2-}$ and alkalinity were stored at 4ºC. The sediments in the centrifuge tubes were stored in aluminum gas-tight bags under a nitrogen atmosphere at -20ºC.

Sediment slices from the third core were taken at 0.5 cm resolution between depths of 0 to 5 cm. Sediments from each depth were stored for the determination of cable bacterial densities by fixing in 0.5 mL ethanol (>99.8%; molecular biology grade) after homogenization and storage in Eppendorf vials (1 mL) at -20ºC. Samples for extraction of DNA and 16S rRNA sequencing were stored in Eppendorf vials (1 mL) at -80ºC.

**2.3 Porewater analysis**

Porewater $NO_x$ was measured with a GalleryTM Automated Chemistry Analyzer type 861 (Thermo Fisher Scientific, USA). Concentrations of $NH_4^+$ were determined using the indophenol blue method (Solórzano, 1969). Dissolved Mn and Fe in the bottom water and dissolved Mn in the porewater was determined by Inductively Coupled Plasma Optical Emission Spectroscopy (ICP-OES; Perkin Elmer Avio 500). The recovery (accuracy) based on the yield of internal standards and spiking

was 106% for Mn and 105% for Fe. Porewater Fe data are from Bruggmann et al. (2023) as determined for sediment cores from the same cast. Porewater $SO_4^{2-}$ was determined with ion chromatography (detection limit < 50 µmol $L^{-1}$). Porewater $H_2S$ was measured spectrophotometrically using the phenylenediamine and ferric chloride method (Cline, 1969; detection limit ~ 1 µmol $L^{-1}$). Alkalinity was measured through titration with 0.01 M HCl within 24 h of sampling.

**2.4 Solid phase analysis**

Porosity was calculated from the weight loss upon drying of the samples in an oven for a week at 60ºC assuming a sediment density of 2.65 g $cm^{-3}$ (Burdige, 2006). Sediments for geochemical analyses were freeze-dried and homogenized and ground under a nitrogen atmosphere to prevent oxidation artefacts (Kraal et al., 2009). Total element concentrations were determined by digesting aliquots of ~125 mg of freeze-dried sediment in a mixture of three strong acids ($HF/HClO_4/HNO_3$), as described

by van Helmond et al. (2018). The residual gel was redissolved in 1 M $HNO_3$ and analyzed for S, Mn, Fe and Al using ICP-OES. The recovery (accuracy) based on the yield of internal standards and spiking for S, Mn, Fe and Al was 98 %, 98 %, 99 % and 101 %, respectively, and the average analytical uncertainty based on duplicates (n = 11) was 2.7 %, 1.9 %, 1.6 % and



1.5 %, respectively. Aliquots of ~300 mg of sediment were decalcified using 1 M HCl (van Santvoort et al., 2002). After drying and re-powdering, the decalcified sediment was subsequently analyzed for its carbon content using a Fisons Instruments

NA 1500 NCS analyzer. Accuracy and precision of analyses were determined based on measurements of the soil standard IVA2, with an internationally certified value of 0.732 wt.% C, after every 10 samples and at the beginning and end of the sequence. The mean value for IVA2 (n = 17) was 0.721 wt.% C, with a standard deviation of 0.006 wt.% C. The analytical uncertainty based on duplicates (n = 13) was 0.12 wt.% for $C_{org}$.

The Fe speciation in the sediment was determined using a sequential extraction procedure (Kraal et al., 2017,

combining steps from Claff et al. (2010) and Poulton and Canfield (2005)) applied to aliquots of ~50 to 100 mg of freeze-dried sediment. This method allows the separation of (1) Fe(II) minerals such as $FeCO_3$ and FeS and easily reducible Fe oxides through a 4h extraction with 1 M HCl and subsequent determination of dissolved Fe(II) and the sum of dissolved Fe(II) and Fe(III), (2) crystalline Fe oxides using a 4h extraction with 0.35 M acetic acid, 0.2 M Na citrate and 50 g $L^{-1}$ Na dithionite, pH 4.8, (3) recalcitrant Fe oxides using 0.2 M ammonium oxalate and 0.17 M oxalic acid and (4) pyrite ($FeS_2$) using an extraction

with 65% $HNO_3$ for 2h. The solutions were analyzed for Fe using the phenanthroline method (Eaton and Franson, 2005). Average analytical uncertainty based on duplicates (total n = 12) was between 1.5 µmol $g^{-1}$ (recalcitrant Fe oxides) and 10.2 µmol $g^{-1}$ (crystalline Fe oxides) for the different fractions.

A range of parameters relevant to Fe and S cycling were determined. Average ratios of Fe/Al (wt%/wt%) were calculated per site to assess the potential enrichment of the sediment with Fe relative to a detrital background (Raiswell et al.,

2018). Taking the sum of all Fe fractions from the Fe speciation as a measure of highly reactive Fe ($Fe_{HR}$), we calculated the average fraction of the total Fe per site that is highly reactive ($Fe_{HR}/Fe_{total}$) and its pyritized fraction ($Fe_{PYR}/Fe_{HR}$) (Poulton and Canfield, 2011; Raiswell et al., 2018). Ratios of Fe/S were determined as a measure of the degree of sulfidization of the Fe, with lower Fe/S ratios pointing to higher degrees of sulfidization (e.g. Kraal et al., 2017).

**2.5. Cable bacteria density**

Cable bacteria were enumerated through fluorescence in-situ hybridization (FISH) as described in detail by Malkin et al. (2022). Cells were detached from sediment particles using acetate buffer to dissolve carbonates, repeated washes with NaCl solution, followed by resuspension in a detergent (Tween 80) and 10% methanol solution and vortexing for 60 minutes. Density centrifugation with 50% (w/v) Nycodenz solution was subsequently used to separate detached microbial cells from sediment

particles (Kallmeyer et al., 2008). Cells were collected from the aqueous layer and captured on a cellulose acetate filter (0.2 µm), of which sections were subjected to staining. Cable bacteria were identified with fluorescence in situ hybridization (FISH) using the rRNA-targeted oligonucleotide probe DSB706 (Manz et al., 1992), which effectively targets cable bacteria (Pfeffer et al., 2012; Schauer et al., 2014). Staining procedures for FISH followed standard protocols (Pernthaler et al., 2001) using 45% formamide for DSB706. After the identity of cable bacteria was confirmed using FISH on a subset of samples,

enumeration proceeded following staining by Sybr Green I. Cells were counted by epifluorescence microscopy, using a Zeiss



Axio Imager 2, equipped with a Photometrics CoolSnap HQ2 camera. A minimum of 200 randomly selected fields were viewed at 630X magnification and the length of each filament encountered was measured within the field.

### 2.6 16S rRNA gene amplicon sequencing

DNA was extracted from thawed sediment aliquots from the upper 5 cm of the sediment (0.5 cm depth resolution) at all sites, using the procedure described by Kim et al. (2023). Following the protocol outlined by the Earth Microbiome Project, the V4-V5 hypervariable region of the 16S rRNA gene was amplified using the modified primer pair 515F (GTG YCA GCM GCC GCG GTA A) and 926R (CCG YCA ATT YMT TTR AGT TT) with partial Illumina adaptors (Parada et al., 2015). Purified PCR products were assessed by gel electrophoresis (1.5% agarose gel stained with ethidium bromide) to verify the presence

of a single target band (~ 485 bp) and an absence of contamination, quantified using a Qubit 2.0 fluorometer and dsDNA BR Assay Kit (Invitrogen), and sequenced on an Illumina MiSeq platform (2 x 300 nt paired-end reads) at GENEWIZ (South Plainfield, NJ, USA). Bioinformatic analysis was performed in R (v 4.3) using the dada2 pipeline (Callahan et al., 2016), and the resultant amplicon sequence variants (ASVs) were taxonomically assigned using the "*assignTaxonomy*" function which uses a naive Bayesian classifier with minimum bootstrap confidence of 80 for genus-level assignments, using the SILVA

ribosomal small subunit database (v138; Pruesse et al., 2007). A maximum likelihood phylogenetic tree of cable bacteria was constructed using RAxML implementing the GTRCAT approximation with 1000 bootstraps. The reference tree was constructed using full length 16S rRNA gene sequences archived in the NCBI nucleotide database. Representatives of all known cable bacteria species reported to date were included (Geelhoed et al., 2023; Plum-Jensen et al., 2024). In the case where multiple metagenomes or whole length 16S gene sequences for the same species have been archived in NCBI, either

the type specimen (Plum Jensen et al., 2024) or the first listing was included. Environmental sequences from this study were placed on the reference tree using the Evolutionary Placement Algorithm (EPA-ng; Barbera et al., 2019) and visualized with FigTree (1.4.3; Rambout Github Repository; Rambout, 2016).

## 3 Results

### 3.1 Bottom water and porewater characteristics


Bottom water $O_2$ concentrations at our study sites ranged from 0 to 50 µmol $L^{-1}$, with the values from the CTD profiling generally equal to or slightly higher than those from the microelectrode measurements (Table 1; Figure 2 and S1). Overall, the values confirm the trend of decreasing bottom water oxygen concentrations with decreasing water depth, in accordance with prior work (Table 1; McManus et al., 2006). No porewater $H_2S$ was detected with the microelectrodes in the upper 3 to 5 cm

at any of the sites (Figure 2). Microelectrode profiles showed a distinct pH minimum in the upper 0.5 to 1 cm of the sediment at the San Clemente, Catalina and San Pedro sites and a more gradual decrease in pH with depth at the Soledad and San Blas sites (Figure 2). The EP profiles for all sites showed no change or a slight decrease with depth in the sediment.





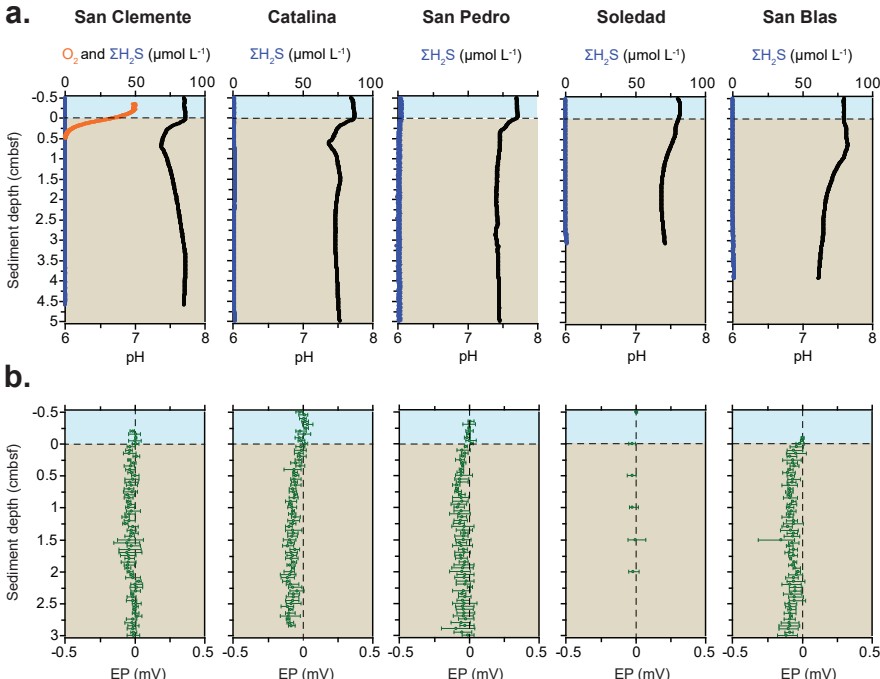

**Figure 2. Micro-electrode profiles of (a) O₂, H₂S and pH and (b) Electric Potential (EP). No accurate O₂ profiles for the Catalina and San Pedro sites are available but O₂ was present in the bottom water (Table 1; see text). The bottom water at the Soledad and San Blas sites was anoxic.**

$NO_x$ was present in the bottom water at all sites, with concentrations decreasing with decreasing water depth from a maximum of 45.2 μmol $L^{-1}$ at San Clemente to a minimum of 23.5 μmol $L^{-1}$ at San Blas (Figure 3). The penetration depths of $NO_x$ showed the same depth trend, i.e. values were lowest at stations with sediments in the ODZ. Distinct maxima in porewater Mn, up to concentrations of ~200 and 25 μmol $L^{-1}$, were observed at the San Clemente and Catalina sites, respectively. At the other sites, porewater Mn remained below 2 μmol $L^{-1}$. Dissolved Fe was present in the porewater at all sites, with concentrations ranging up to 87 μmol $L^{-1}$. Porewater Fe was lowest in the oxic surface sediment of the San Clemente site and below 5 cm depth at the Soledad site. Porewater $SO_4^{2-}$ showed little change with depth at the San Clemente and Catalina sites. A decrease of porewater $SO_4^{2-}$ with depth of between 3.4 and 7.5 mmol $L^{-1}$ was observed at the other sites, however, with the strongest decrease at the Soledad site. Porewater $H_2S$ was generally below the detection limit at all sites except for Soledad. Here, $H_2S$ emerged in the porewater around a depth of 6 cm and reached a maximum value of 485 μmol $L^{-1}$ at 15 cm depth. Porewater $NH_4^+$ increased with sediment depth at all sites. Maximum values of $NH_4^+$ in the porewater of 350, 62 and 271 μmol $L^{-1}$ were observed for the San Clemente, Catalina and San Pedro sites, respectively. Much higher maximum values of 1980 and 886 μmol $L^{-1}$ were observed for the Soledad and San Blas sites, respectively. Alkalinity ranged between 2.3 and 13 mmol $L^{-1}$ and the trends with depth and between stations largely followed those of $NH_4^+$. Porosity decreased with sediment depth at all sites (Table S1).



**Figure 3. Porewater profiles of NO$_x$, dissolved Mn, Fe$^{2+}$, SO$_4^{2-}$, H$_2$S, NH$_4^+$ and alkalinity for the five sample locations. Note the differences in the concentration scales for dissolved Mn, NH$_4^+$ and alkalinity.**




## 3.2 Solid phase profiles

Sediment $C_{org}$ contents ranged between 2.9 and 8.2 wt%, with the lowest and highest values found at the San Clemente and Soledad sites, respectively (Figure 4). At each site, relatively little change in $C_{org}$ with sediment depth was observed. Total S
contents ranged between 95 and 350 µmol g$^{-1}$ and decreased with depth at all sites, except at the Soledad site. Here, quite some variability and an overall increase in S with depth was observed. Surface enrichments of Mn were observed at the San Clemente and Catalina sites with maximum concentrations of ~1200 and 15 µmol g$^{-1}$, respectively. At the other sites, Mn concentrations changed little with depth and were low, ranging between 7 to 9 µmol g$^{-1}$ at San Pedro and 3 to 5 µmol g$^{-1}$ at the Soledad and San Blas sites. Trends in Mn/Al profiles closely followed those of Mn. Total Fe contents showed variable trends with depth at
the various stations, with the highest values observed at the San Pedro site and lowest at the Soledad site. Ratios of Fe/Al mostly ranged between 0.6 and 0.8 (wt%/wt%), except for the San Blas site where ratios were mostly around 0.4 (wt%/wt%). Surface enrichments in easily reducible Fe oxides (i.e. HCl-extractable Fe(III)) were observed at all sites, with the highest maximum value of 412 µmol g$^{-1}$ at the San Pedro site and the lowest maximum value of 48 µmol g$^{-1}$ at the Soledad site.

An increase with depth of FeS and FeCO$_3$ (i.e. HCl-extractable Fe(II)) over the upper ~5 cm was observed at all sites except San Blas, where a small decrease with depth was seen. Sediment pyrite concentrations ranged from 2 to 60 µmol g$^{-1}$ and increased with depth in the upper few cm's of the sediment at all sites, except for San Pedro, where values remained constant. Crystalline and recalcitrant Fe oxides showed variable trends with depth and between stations (Table S1).

Average weight ratios of Fe/Al per site ranged from 0.44 to 0.74, with the lowest values observed at the Soledad and
San Blas sites (Table 2). Ratios of FeHR/Fetotal range from 0.32 to 0.39, without a distinct trend between sites. Ratios of FePYR/FeHR ranged from 0.10 to 0.29, with the lowest value for the San Clemente site and highest one for Soledad. Molar Fe/S ratios ranged from 10.7 at San Clemente to 1.7 at the Soledad site.

**Table 2. Average ratios of Fe/Al (wt% wt%$^{-1}$), FeHR/Fetotal, FePYR/FeHR and Fe/S (mol mol$^{-1}$) for the sediments at**
**all sites.**

| Site | Fe/Al (wt% wt%$^{-1}$) | FeHR/Fetotal | FePYR/FeHR | Fe/S (mol mol$^{-1}$) |
|---|---|---|---|---|
| San Clemente | 0.74 | 0.39 | 0.10 | 10.7 |
| Catalina | 0.67 | 0.32 | 0.14 | 6.4 |
| San Pedro | 0.74 | 0.37 | 0.13 | 6.2 |
| Soledad | 0.61 | 0.32 | 0.29 | 1.7 |
| San Blas | 0.44 | 0.36 | 0.10 | 3.2 |

## 3.3 Cable bacteria density (microscopy results)

The volumetric length density of cable bacteria (m cm$^{-3}$) as determined by microscopy and DSB-FISH probes for the upper 2.5 cm of the sediment varied strongly between sites and with depth in the sediment (Figure 5). Cable bacteria densities were



lowest at the San Clemente site (maximum of 0.4 m cm$^{-3}$ in the depth interval from 0.5 to 1.0 cm) and were highest at the Soledad site (maximum of 157 m cm$^{-3}$ in the depth interval from 0 to 0.5 cm). Almost no cable bacteria filaments were observed below a depth of 1.5 cm. Areal length densities (m cm$^{-2}$) showed an increasing trend in the following sequence: San Clemente (0.2 m cm$^{-2}$), Catalina (9.4 m cm$^{-2}$), San Pedro (16.2 m cm$^{-2}$), San Blas (19 m cm$^{-2}$) and Soledad (129 m cm$^{-2}$).

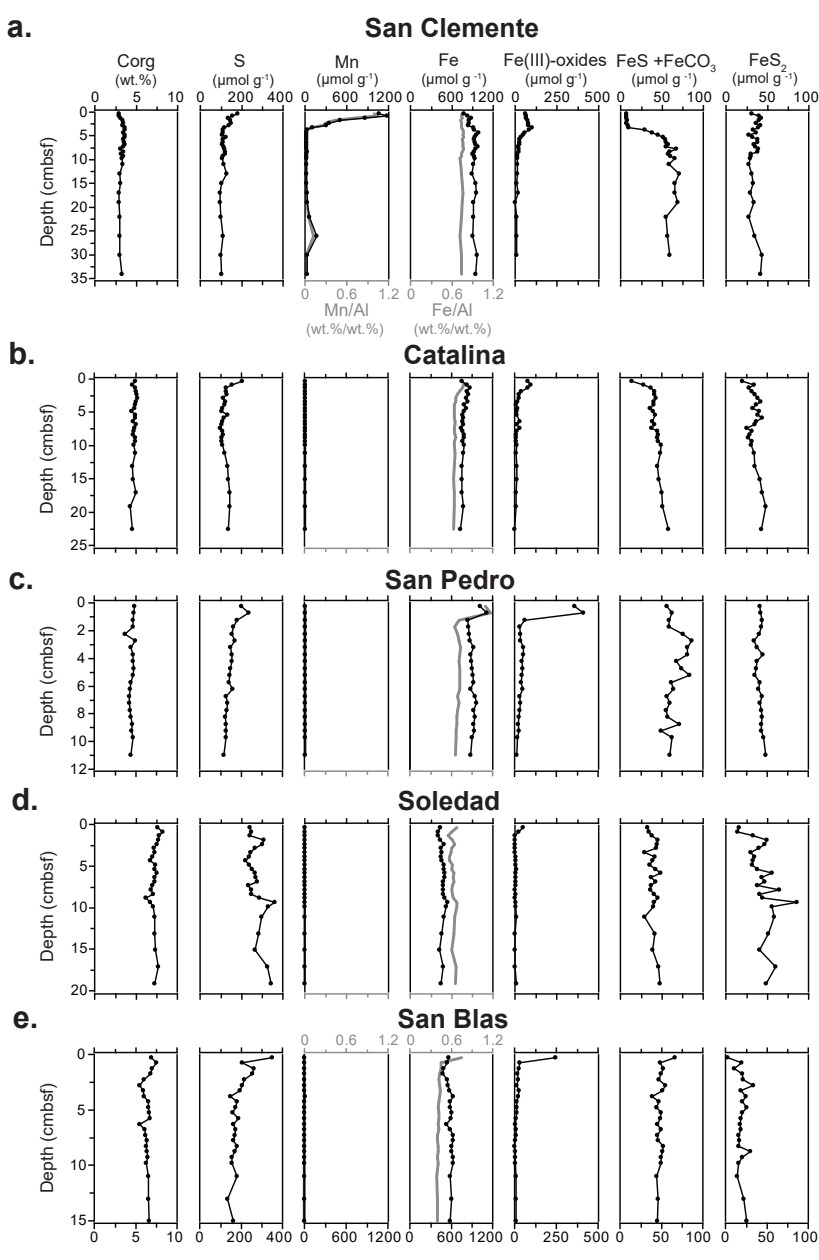


**Figure 4. Sediment profiles of $C_{org}$, S, Mn, total Fe and Fe/Al, Fe oxides (Fe(III) oxides, HCl-Fe(III)), FeS and FeCO₃ (HCl-Fe(II) and FeS₂ (HNO₃-Fe), for all five sample locations.**





**3.4 16S rRNA amplicon sequencing**

Differences in microbial community composition between sites (i.e., beta diversity) and with depth were evident in the NMDS ordination (Figure 6 and S3). Measured to 5 cm depth, community composition changed more rapidly with depth at the hypoxic sites compared to the anoxic sites.

Key drivers of the inter-site community differences included higher relative abundances of Alphaproteobacteria (class), Gemmatimonadota (phylum) and Nitrospirota (phylum) at the hypoxic sites, while Bacteroidota, Desulfobacterota,
and Sva0485 (phyla) were more abundant at the anoxic sites. Near the sediment surface, *Gammaproteobacteria* were more abundant at hypoxic sites than at anoxic sites. *Firmicutes* exhibited the highest relative abundance at San Pedro and San Blas but contributed a smaller fraction of the community at San Clemente and Catalina. Notably, the site with the highest bottom-water oxygen (San Clemente) contained a uniquely high proportion (11.4%) of the archaeal class *Nitrososphaeria* in surface sediments. The greatest proportion of Cyanobacteria was observed at San Blas (up to 4.6 % of the community), followed by
the upper 2.5 cm sediments at Soledad and San Pedro sites.

Alpha diversity (i.e., within-sample diversity) generally decreased with depth (Figure S4). This decline was most pronounced at San Clemente, while San Blas showed no appreciable decrease. The Shannon diversity index exhibited the clearest depth-related patterns, whereas the Chao1 and Faith's phylogenetic diversity indices showed more variability, potentially reflecting biases from uneven sequencing depth.

Cable bacteria reads exhibited low relative abundance at all sites (<0.05 % of the total reads). Nevertheless, multiple amplicon sequence variants (ASVs) affiliated with cable bacteria were identified across the three sites with the lowest oxygen concentrations: San Pedro (2 ASVs), Soledad (6 ASVs), and San Blas (14 ASVs). To explore evolutionary relationships, recovered cable bacteria ASVs were mapped onto a phylogenetic tree constructed using NCBI-archived 16S rRNA gene sequences (Figure 7). At the hypoxic San Pedro site, the two recovered ASVs clustered within a poorly resolved clade that
included the following species: *Ca*. E. japonica, *E. communis RB*, *E. rattekaaiensis Rat3*, *Ca.* E. marina, and *E. arhusiensis* (following proposed nomenclature and type species identification from Plum-Jensen et al. (2024)). An additional four ASVs from the Soledad or San Blas were distributed throughout this unresolved clade. The remaining cable bacteria ASVs originated from Soledad and San Blas and clustered with three other groups: Four ASVs from San Blas clustered with *Ca.* Electrothrix sp. EH2, five ASVs from both anoxic basins affiliated with *E. gigas*, and five ASVs, primarily from San Blas, clustered with
the deeper-branching undescribed genus AR3.




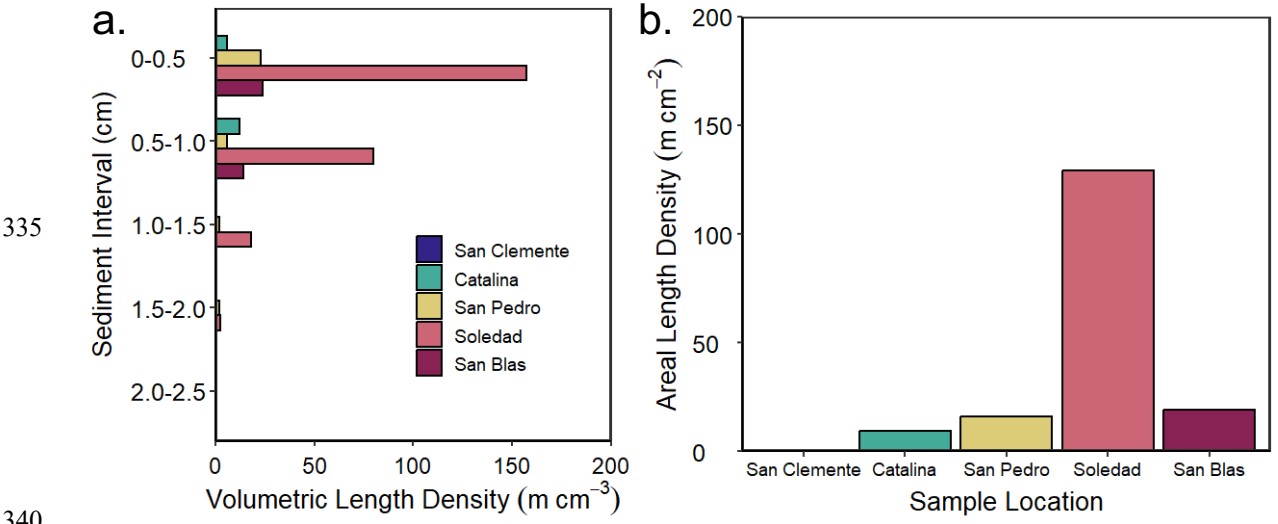


**Figure 5. Cable bacteria densities for the upper 2.5 cm of the sediment at all five sample locations. DSB indicates cable bacteria, as enumerated with the DSB706 oligoprobe (a) Volumetric length density in m cm$^{-3}$, (b) Areal length density in m cm$^{-2}$.**

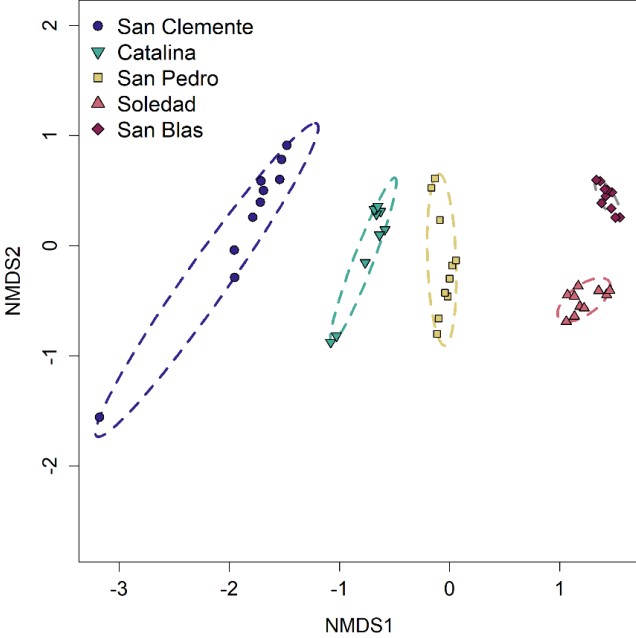

**Figure 6. Beta Diversity. Non-metric multidimensional scaling (NMDS) ordination on Bray-Curtis dissimilarity distances, with**
**ellipses indicating sites. Arrows have been added to indicate direction of increasing depth below the seafloor. Stress = 0.062.**



**Fig. 7. Phylogenetic tree of the known cable bacteria clades based on near full length 16S rRNA gene sequences constructed using maximum likelihood (RAxML) and a GRTCAT approximation. The cable bacteria reference tree is comprised of sequences reported in primary literature (Trojan et al., 2016; Geelhoed et al., 2023; Sereika et al., 2023; and Plum-Jensen et al., 2024), using accession numbers therein. The positions of the environmental amplicons observed in this study are shown in color and indicated by the prefix "WCT" indicating the Pacific transect, and the most abundant amplicon from Chesapeake Bay is indicated with the prefix "CB". Bootstrap support greater than 70 (1000 iterations) of the reference tree is indicated by a filled circle at the node.**





## 4 Discussion

### 4.1. Cable bacteria density and activity

Our results reveal the presence of cable bacteria in ODZ sediments along the Californian and Mexican margin. The cable
bacteria densities observed here (0.2 to 129 m cm$^{-2}$) are low when compared to *in-situ* abundances in coastal sediments with relatively oxygen-rich bottom waters (>100-350 µmol L$^{-1}$ O$_2$). Examples include sediments in the Den Osse basin of seasonally hypoxic marine Lake Grevelingen (402-408 m cm$^{-2}$; Seitaj et al., 2015), muddy sediments in the North Sea (168-352 cm$^{-2}$; van de Velde et al., 2016) and sediments with seasonally anoxic bottom waters in Chesapeake Bay (174-447 m cm$^{-2}$; Malkin et al., 2022). The cable bacteria densities in ODZ sediments are comparable, however, to those observed in a variety of hypoxic and
anoxic environments in the Baltic Sea (0-150 m cm$^{-2}$; Marzocchi et al., 2018; Hermans et al., 2019). These settings mostly had oxygen concentrations below 50 µmol L$^{-1}$, i.e. the range was comparable to that of our sites (Table 1). Strikingly, we see the highest abundance of cable bacteria at the anoxic Soledad site and the lowest abundance at the hypoxic San Clemente site (Figure 5), i.e. the opposite of what would be expected based on oxygen availability.

       We find no evidence for cable bacteria activity at the time of sampling. The distinct pH maximum near the sediment-
water interface that is characteristic for their activity was not present and the relatively limited acidification of the porewater with depth (Figure 2) suggests that other processes were responsible for the pH trends (e.g. nitrification; Silburn et al., 2017). The lack of activity based on the pH profiles was corroborated by the lack of a rise in electric potential with depth at all sites, which would be expected if there was electron transport through the cables (Damgaard et al., 2014). Low oxygen is known to negatively affect cable bacteria activities and densities (Burdorf et al., 2018; Marzocchi et al., 2018; Hermans et al., 2019) but,
as shown by Marzocchi et al. (2018) for the Baltic Sea, this does not preclude cable bacteria activity, even when O$_2$ concentrations are as low as 5 µmol L$^{-1}$. In the next sections, we will discuss potential explanations for our observations, primarily related to the electron donor and electron acceptor supply.

### 4.2. Electron donor and acceptor supply

Cable bacteria require H$_2$S for their activity (Pfeffer et al., 2012). At our study sites, the cable bacteria are mostly concentrated in the upper 1 to 1.5 cm of the sediment near the sediment-water interface (Figure 5). This implies that the H$_2$S should be supplied to this upper sediment layer. There are three potential sources of H$_2$S in a setting like this (Risgaard-Petersen et al., 2012; Hermans et al., 2019): upward diffusion of H$_2$S from deeper layers, sulfate reduction in the zone where the cable bacteria are active and dissolution of FeS. At the time of sampling, H$_2$S was either absent from the porewater or, as was the case at the
Soledad site, emerged below a depth of 6 cm, implying that upward diffusion of H$_2$S can be excluded as a source. Direct supply of H$_2$S through sulfate reduction at the time of sampling is likely also limited: easily reducible Fe oxides were consistently present in the upper cm's of the sediment (Figure 3) and are thermodynamically preferred over sulfate as an electron acceptor in organic matter degradation (Froelich et al., 1979). The role of FeS is difficult to evaluate because we only have a measure



of the sum of FeS and FeCO₃. The sum of both fractions (Figure 4) shows a slight depletion in the surface sediments at all

sites except San Blas, suggesting that dissolution of FeS is a possibility. Since the FeS also would have to be formed in the surface sediment, this implies that, for FeS to be a source, redox conditions in the sediment at this site would have to be variable, i.e. a period of buildup of FeS would have to be followed by a period of FeS loss, as observed in seasonally hypoxic systems (e.g. Seitaj et al., 2015; Malkin et al., 2022). During such periods, sulfate reduction could also directly fuel the activity of cable bacteria.

Seasonal and interannual variability of primary productivity and, hence, of organic matter supply to the sediment is a common feature of the basins along the Californian and Mexican margin (Thunell, 1998; Silverberg et al., 2004; Collins et al., 2011). This is also reflected in the $C_{org}$ profiles for our sites, which show quite some variability with sediment depth, especially at the anoxic Soledad and San Blas sites. Such $C_{org}$ trends deviate from the typical downward decay profiles observed in sediments with a constant input of organic matter (Burdige, 2006). The spikes in total S suggest variations in the in-situ

production of $H_2S$ and its conversion to Fe sulfides, in line with temporal variability in the input of organic matter (Figure 4). Seasonal increases in input of organic matter have been shown to stimulate sulfate reduction in surface sediments (e.g. Thamdrup et al., 1994) and such pulsed inputs may supply $H_2S$ to cable bacteria (Malkin et al., 2022). We find a positive relationship between the burial of $C_{org}$ and cable bacteria density, supporting a role for increased organic matter input (Figure 5; Table 1). We also find the highest cable bacteria densities for sediments with the lowest Fe/S ratios (Figure 5; Table 2), i.e.

for sites where there is a higher supply of S relative to Fe, supporting a role for periodic enhanced sulfate reduction.

The cable bacteria also require an electron acceptor. To date, only oxygen, nitrate and nitrite are known to act as electron acceptors for cable bacteria (Pfeffer et al., 2012; Marzocchi et al., 2014), although solid phase electrodes have been suggested (Reimers et al., 2017; Li et al., 2020; Bonné et al., 2024). Increased coastal upwelling in the region in winter and spring not only enhances primary productivity but also can lead to periodic, sometimes even seasonal inflows of oxygen into

the basins, as reported for the Santa Barbara basin (e.g. Bograd et al., 2002; Peng et al., 2024). The frequency of reoxygenation of the basins studied here is less well known, especially for the Soledad and San Blas basins. The relatively low ratios of FeHR/Fetotal point towards a setting that is "possibly anoxic" (Poulton and Canfield, 2011) and hence does not exclude periodic reoxygenation. The relatively low degree of pyritization of the Fe (FePYR/FeHR <0.3) is consistent with such an environment (Poulton and Canfield, 2011; Raiswell et al., 2018). The Fe/Al ratios are variable and do not provide additional

constraints in this setting.

Nitrate is always present in the bottom water at our study sites (Table 1), indicating that an electron acceptor for the cable bacteria is present. Nitrate is a much less effective electron acceptor than oxygen but is expected to be used when oxygen is absent (Marzocchi et al., 2014). One could expect that cable bacteria in an anoxic, nitrogenous setting such as that of San Blas and Soledad would be adapted to using nitrate. However, given the lack of a signal for cable bacteria activity in the

profiles of pH and electric potential in the presence of nitrate, we conclude that nitrate was not being used at the time of sampling and hence, was not limiting. Instead, the limiting factor for cable bacteria activity in these sediments was likely the



electron donor supply, i.e. the availability of $H_2S$. This availability likely changes over the season and is expected to be highest during periods of high productivity upon coastal upwelling, possibly combined with periodic reoxygenation.

**4.3 Patterns in the microbial community composition**

The microbial communities in the sediments of these low oxygen basins exhibited highly structured patterns of alpha (within-sample) and beta (between-sample) diversity (Figure S4 and 6, respectively). In these hypoxic and anoxic bottom water conditions, sediment mixing by meso- and macrofauna is restricted or absent (Levin, 2003), which contributes to the pronounced depth-structured patterns observed. Alpha diversity generally declined with depth (examined to 5 cm) at all sites except San Blas, consistent with sediments functioning primarily as a filter, with new species arriving at the sediment surface, and clades lost over time with progressive depth (Starnawski et al., 2017; Kirkpatrick et al., 2019). Variations in community structure across the study sites were likely strongly influenced by bottom water oxygen levels and sedimentation rates, which are well-documented as dominant forcing factors (Hoshino et al., 2020).

Distinct patterns emerged in the microbial community data, from which some biogeochemical inferences can be made. At San Clemente, the site with the highest bottom water oxygen concentration, surface sediments harbored a uniquely high proportion of the Archaeal class *Nitrososphaeria* (~11 %; Figure S3), which are typified by an ability to perform ammonia oxidation (Könneke et al., 2005, Stieglmeier et al., 2014), the rate limiting step in marine nitrification. Additionally, all three hypoxic sites supported modest abundances of the phylum Nitrospirota (primarily *Nitrospira),* a bacterial phylum known for diverse metabolic abilities, canonically including nitrite oxidation (D'Angelo et al., 2023). In contrast, these taxa were nearly absent at the anoxic Soledad and San Blas sites, where anaerobic phyla typical of reducing sediments were enriched. These included Bacteroidota, which have vast polymer degradation ability and are considered the primary degraders of organic carbon in marine settings (Fernández-Gómez et al., 2013); Desulfobacterota, the dominant phylum responsible for sulfate reduction in marine sediments (Wasmund et al., 2017); and Sva0485, a lesser understood taxon, potentially involved in sulfate reduction, sulfide oxidation, and iron metabolism (Tan et al., 2019). These patterns suggest that oxygen availability, particularly at San Clemente, support significant sediment nitrification capacity, compared with the anoxic sites, which appear adapted to more reducing conditions in the upper 5 cm. These findings reinforce the idea (Section 2.1) that the bottom-water oxygen concentrations during our sampling campaign are representative of persistent differences between sites.

Cyanobacteria were more prevalent at the most hypoxic and the anoxic sites. These bacteria likely represent organic material originating from the upper ocean euphotic zone. Their accumulation is likely indicative of slower rates of degradation in the sediments in the absence of oxygen (Hartnett and Devol 2003; Ruvalcaba Baroni et al., 2020).

Amplicon sequence variants (ASVs) affiliated with cable bacteria were only identified at the three sites with the lowest oxygen concentrations (Figure 7). Visual observations at these sites did not point to the presence of *Thioploca* or other giant sulfur-oxidizing bacteria. Because of introns in their ribosomal genes, their presence cannot be reliably assessed based on 16S rRNA amplicon sequencing (Salman et al., 2012). *Thioploca* has been observed in sediments of Soledad basin (Chong et al., 2012), previously, however, suggesting the possibility of an overlapping niche with cable bacteria, as observed in coastal





sediments (Seitaj et al., 2015; Malkin et al., 2022). Expansive *Thioploca* mats have also been reported from the nearby oxygen-depleted Santa Barbara Basin, where they are primarily confined to the depocenter, a region characterized by the presence of nitrate and absence of oxygen (Yousavich et al., 2024). Although cable bacteria were not examined in that study, it was hypothesized that they dominate shallower oxygenated depths, while *Thioploca* occupies deeper anoxic zones. However, the

presence of cable bacteria in the anoxic Soledad Basin challenges these assumptions, suggesting the possibility cable bacteria may be adapted to nitrate-based respiration, and that their niche partitioning may be driven by different factors. Further studies of the spatial distribution of cable bacteria, their activity and that of other sulfur-oxidizers in sediments of ODZs will be critical to unravelling the ecological interactions and metabolic activities of this microbial group.

**4.4 Diversity of cable bacteria**

Mapping recovered cable bacteria ASVs onto a phylogenetic tree provided insights into their distribution across these low oxygen basins (Figure 7). The reference tree largely mirrored the genus-level architecture of other more extensive phylogenetic or phylogenomic trees (Plum-Jensen et al., 2024; Geelhoed et al., 2023), though species-level topology and genus-level diversity are still being refined (Ley et al., 2024). Cable bacteria ASVs were detected exclusively in the basins

with the lowest oxygen concentrations (San Pedro, Soledad, and San Blas). Most ASVs were distributed across the known diversity of the genus *Ca*. Electrothrix, with none affiliated with the freshwater genus *Ca*. Electronema or its low salinity-tolerant sister clade represented by *Electronema halotolerans* (Trojan et al., 2016, Sereika et al., 2023, Plum-Jensen et al., 2023, Ley et al., 2024). These results support the understanding that salinity plays a key role in the phylogeography of cable bacteria (Dam et al., 2021; Ley et al., 2024). Our results also affirm the frequent observation that multiple cable bacteria species

commonly co-occur in the same setting (Malkin et al., 2022, Ley et al., 2024), yet the distinct selective advantages that enable their coexistence remain poorly understood.

The phylogenetic analysis highlighted two notable clusters. The first included ASVs affiliated with *Ca. Electrothrix gigas*, a recently described species characterized by unusually large cell diameters up to 8 µm (Geelhoed et al., 2023), compared to the typical 0.6–2 µm range reported for other cable bacteria (Schauer et al., 2014; Malkin et al., 2014). The propensity for

'gigantism' in *E. gigas* appears to have a genetic basis linked to an actin-like 'big bacteria protein', encoded by *bbp*, which may have originated in *E. gigas* through horizontal gene transfer from *Thioploca* or a close relative (Geelhoed et al., 2023). While small cell size is advantageous in substrate-limited environments due to a higher surface area-to-volume ratio, gigantism occurs across diverse microbial lineages and is often associated with storage inclusions that reduce cytoplasmic volume and enable substrate or product storage (Schulz and Jørgensen, 2001; Jørgensen, 2010). For *Thioploca*, large size supports nitrate

storage and access to deeper sulfide. However, *E. gigas* achieves its size without known storage mechanisms, highlighting a key knowledge gap in its ecology and evolutionary drivers. *Ca*. E. gigas was previously collected from intertidal sediments (Rattekaai, Netherlands; Geelhoed et al., 2023) and this is, to our knowledge, the first time it is reported for nitrogenous basins. Its presence in these sediments suggests that these habitats may be particularly well suited for driving large cell size, though the specific selective pressures remain unclear.



The second notable cluster included five ASVs, primarily from San Blas, affiliated with the undescribed genus AR3. Current genomic insights into this genus derive from a single filament collected from cohesive sediments in the North Sea (Station 130, Belgium), where persistent cable bacteria populations have been linked to anthropogenic disturbances (Malkin et al., 2014; van de Velde et al., 2016). While environmental sequences affiliated with *AR3* have been detected in the Baltic Sea (Dam et al., 2021), and in association with roots of cosmopolitan seagrass species (Scholz et al., 2021), a recent exploration

of cable bacteria phylogeography suggested this putative genus may be particularly well adapted to deep sea sediments, with most occurrences in long read archives coming from deep sea samples, including hydrothermal vents and mud volcanoes (Ley et al., 2024). Our study adds a new dimension to this enigmatic genus's distribution by identifying its occurrence in an anoxic, nitrogenous basin, raising the possibility that members of this clade may support relict or novel adaptations worthy of further exploration, particularly with regard to nitrate respiration.

Laboratory and genomic analyses demonstrate that cable bacteria can perform dissimilatory nitrate reduction to ammonium (DNRA), likely serving to dissipate electrons during sulfide oxidation-based energy acquisition (Marzocchi et al., 2014, 2022; Kjeldsen et al., 2019). This ability to respire nitrate was likely acquired through multiple horizontal gene transfer events that predate both their ability to use oxygen as an electron acceptor and the divergence of *Electrothrix* and *Electronema* (Marzocchi et al., 2022). Notably, this pathway was found complete in *Electronema* but not in *Electrothrix* (Marzocchi et al.,

2022). Examination of AR3 genomes and enrichment cultures from nitrogenous basins may reveal whether these clades are particularly adapted to nitrate respiration or if, like coastal *Electrothrix*, they exhibit lower electroactivity when coupling sulfide oxidation with nitrate reduction rather than oxygen respiration, indicating lower efficiency of the nitrate respiration pathway in extant coastal *Electrothrix* (Marzocchi et al., 2014).

**4.5 Outlook**

      Cable bacteria are distinguished from their sulfate reducing or sulfur disproportionating relatives by an unprecedented long distance electroactive lifestyle that enables an energy-conserving sulfide oxidation metabolism at anoxic sediment depths. Cable bacteria possess numerous evolutionary innovations that have enabled this unique lifestyle, positioning them as important catalysts of biogeochemical cycling (e.g. Risgaard-Petersen et al., 2012; Seitaj et al., 2015; Sulu-Gambari et al.,

2016). The hallmark of cable bacteria is their periplasmic electron conductor—a novel, yet incompletely understood, feature (Boschker et al., 2021). This electroactive capacity relies on co-evolved traits, evidenced by hundreds of unique genes absent in other sulfur-reducing *Desulfobulbaceae* (Kjeldsen et al., 2019). Genomic analyses have identified that approximately half of these unique genes arose from mutations of ancestral genes, while the other half have been acquired through horizontal gene transfer, likely from other sediment-associated bacteria that share an overlapping niche. These co-evolutionary interactions

highlight the importance of studying cable bacteria in anoxic basins, where enigmatic clades like AR3 and potentially *E. gigas* may bridge gaps in understanding their evolutionary history. Future research should prioritize isolating filaments, utilizing longer read sequencing platforms, and deeper metagenomic analyses. Cable bacteria from these basins may be particularly useful for further investigating the evolution of their nitrate and oxygen utilization pathways as well as the ecological drivers

of 'gigantism'. Further work should also focus on elucidating the role of cable bacteria versus other sulfur-oxidizing bacteria

in removing sulfide from continental margin sediments overlain by oxygen-deficient, nitrogenous waters.

## 5. Conclusions

We show evidence of a niche for cable bacteria in continental margin sediments overlain by hypoxic and anoxic bottom waters in basins along the coasts of California and Mexico. The abundance of cable bacteria based on fluorescence in-

situ hybridization ranged from 0.2 to 19 m cm$^{-2}$, with highest values at the sites with anoxic bottom waters. No cable bacteria activity could be detected at any of our study sites. Our results suggest that cable bacteria filaments and their DNA persisted at the sites where oxygen was most limiting following a period of activity. This observation points to the following scenarios: (1) periodic bottom water ventilation events may temporarily provide oxygen to sustain cable bacteria activity, with slower DNA turnover occurring overall at the sites with more persistent anoxia; and/or (2) cable bacteria activity may be fuelled by

periodic elevated inputs of organic matter and associated enhanced sulfate reduction. We note that, in this latter case, cable bacteria in the anoxic basins may rely on nitrate rather than oxygen as a terminal electron acceptor. The observed filament density at Soledad Basin suggests a dominance of nitrate utilization is possible.

Using phylogenetic analysis, we show that the diversity of the amplicon sequence variants (ASVs) was spread across the *Candidatus* Electrothrix lineage, including multiple ASVs related to *Electrothrix gigas*, a species of giant cable bacteria.

Additionally, multiple sequences revealed affiliation with a deeper branching clade sister to *Electrothrix*, which we hypothesize is a novel genus.

## Data availability

The geochemical data presented in the paper are deposited in the Zenodo repository doi.: 10.5281/zenodo.14896362. Amplicon

sequence libraries are publicly available at the NCBI Sequence Read Archive. Submission Number: SUB15094649 and BioProject ID Number: PRJNA1224808

## Conflict of interest

The authors declare that they have no conflict of interest.


## Supplement

The supplement related to this article is available online.

**Author contribution**: CS, MH, SM, JM and SS planned the fieldwork and analyses, MH, JM and SS performed the sampling,

MH, NH, SM performed the chemical and microbial analyses, CS, NH, SM, MH and ME analyzed the data. CS and SM wrote the manuscript draft. All authors reviewed and edited the manuscript.




**Acknowledgments**

We thank the captain and crew of R/V *Oceanus*, Kristin Ungerhofer and Claire Reimers for their support before and/or during the sampling campaign. We are also grateful to M. Mikos, H. de Waard, C. Mulder, O. van Loenen and Z. Wang for analytical assistance at Utrecht University. This research was financially supported by ERC Synergy grant Marix (8540088), the

Netherland Earth System Science Center (NESSC 024002001), the Dutch Research Council (NWO Vici grant 865.13.005) and the National Science Foundation (grants OC-1657832, OC-1657690 and OCE-1756877).



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
