# Peer review of "A niche for diverse cable bacteria in continental margin sediments overlain by oxygen-deficient waters"

_EGUsphere, 2025_

## Author Response (AR1)

Response to reviews #1 and #2

Reviewer #1

General comments:

In this manuscript, Slomp et al. initially described the continental margin sediments (>100 m deep) of hypo- to anoxic nitrogenous basins as a niche for the development of cable bacteria (CB). Then, they estimated the abundance and diversity within the *Cadidatus* Electrothrix lineage to provide an overview of the recently described diversity of this group, and to suggest a new genus adapted to these environmental conditions. The manuscript is fairly well structured, and its objectives are clear. The methodological approaches are comprehensive and aim to characterise the geochemical conditions of the bottom waters, the sedimentary compartment (pore-water and solid phases) and the microbial community concerned. The data sets are freely accessible on Zenodo. Despite the wide range and quality of the data, it is a snapshot in time, which limits the scope of the discussion on the factors controlling the CB dynamics in this potential niche. Overall, I think this is a robust and very interesting research paper which corresponds to the scope of Biogeosciences.

We thank the reviewer for the positive words.

Specific comments:

Line 104: you present bibliographical data on *Thioploca* in the Soledad basin, but nothing on the other groups presented in the introduction (*Beggiatoaceae* and *Thiomargarita*). Is there any information available on this subject in these basins? Similarly, in line 108, you present (admittedly old) data on the meiofauna and macrofauna of this same basin, but you say nothing about the other basins? Have no data been published?

Reply: We would be happy to include such data. However, to our knowledge there are no published studies on *Beggiatoaceae* or *Thiomargarita* for these five basins. The same holds for studies of the meiofauna and macrofauna in the basins other than Soledad.

Lines 133-135: only one $O_2$ profile was done in San Clemente? How were the pH and $H_2S$ measurements set up to achieve complete microprofiling to a depth of over 3 cm in less than 15 minutes? I assume that the waiting and measurement times are particularly short. Is this relevant for this type of measurement?

Reply: Indeed, only one $O_2$ profile was collected in San Clemente basin. The measurement times for the pH and $H_2S$ profiles were kept as short as possible to limit the potential for changes in the profiles following core collection. We have provided more information on this in the revised manuscript. Line numbers in our replies refer to the track changes version of the manuscript.

Revised text:

L148-152:"Profiles of pH and $H_2S$ were measured simultaneously at all sites as quickly as possible to limit potential changes in the profiles. Each profile took 10-15 minutes using a measurement time of 1.5 second at each depth. Because of interference from the ship during sailing, no duplicate profiles could be obtained. We note that pH profiles have a high reproducibility and reporting single profiles, as is done here, is common (e.g. Seitaj et al., 2015; Daviray et al., 2024). Finally, the EP profiles were measured in triplicate for all cores, with each profile taking 3 minutes."

Line 160: one point I don't understand about the analysis of dissolved metals: why didn't you use the pore water Mn data from Bruggmann et al. (2023), as you did for Fe, in order to have the same vertical resolution for both? The vertical resolution in Bruggmann et al. (2023) is low and I find it relevant that you have carried out higher resolution analyses for Mn. But in that case, why didn't you do the same for Fe to get geochemistry within the same core?

Reply: Our study focuses on characterizing the geochemical environment of the cable bacteria in these sediments. This implies that the high resolution porewater data collected through centrifugation as presented in our manuscript are more relevant than the low-resolution profiles collected at the same sites (but from different multicores) through rhizon sampling by Bruggmann et al. (2023). Unfortunately, we do not have Fe data from the ICP-OES for our samples. This is why we chose to include the Bruggmann et al. (2023) Fe data. The main significance of the Fe porewater data is that they show evidence for Fe reduction throughout most of the profile, which is relevant since this is what would be expected in the absence of free sulfide in the porewater. Given that the Bruggmann et al. (2023) Mn profiles generally show similar depth trends and contrasts between stations as our high-resolution data, we do not see the added value of adding those data. The Mn data also are only provided as supporting data here and are not key to the discussion. We have modified the methods section and caption of figure 3 to indicate that the samples of Bruggmann et al. (2023) are from rhizon samplers and have a lower depth resolution.

Revised text:

Materials and methods:

L187: Porewater Fe data are from Bruggmann et al. (2023) as determined for sediment cores from the same cast using rhizon samplers at lower depth resolution.

Caption of Figure 3:

L306: Note that the porewater $Fe^{2+}$ profiles are from Bruggmann et al. (2023) and were obtained with rhizon samplers at a lower depth resolution.

Section 4.1: the densities observed in these hypo-anoxic basins are like those observed *in situ* on estuarine intertidal mudflats (oxygenated environment + sulphides) where CB could be particularly active (Daviray et al., 2024), or in the rhizosphere of aquatic plants (Scholz et al., 2019, 2021).

Reply: We are aware of the reviewer's excellent work on cable bacteria in estuarine intertidal mudflats (Daviray et al., 2024) and that of Scholz et al. (2019; 2021) on cable bacteria in the rhizosphere of plants. We prefer not to include references to the cable bacteria abundances in those studies since those sites are very different from the type of sites studied here. We have added that we are comparing our cable bacteria to "submerged coastal sediments" in the text.

Revised text: L420-422: "We focus here on a comparison to submerged coastal sediments only, given their greater similarity in environmental characteristics to ODZ settings than, for example, intertidal sediments."

The presence of CB (DNA data) but the absence of activity also raises a hypothesis that is not discussed here: could it be the result of CB-enriched sediment transport into the basins? Do you have any information on the marine currents affecting these locations? In this case, it could be better to talk about a 'potential' niche.

Reply: There is no information on the potential occurrence of cable bacteria in the near-coastal regions adjacent to these basins. Importantly, however, these basins are characterized by generally low rates of detrital sediment input (e.g. van Geen et al., 2003) and, due to their offshore position, low rates of sedimentation overall (Table 1). If at Soledad basin, the site with the highest cable bacteria abundance and sedimentation rate, the cable bacteria had been transported long-distance through the water column (there is no evidence that that is possible over such long distances, even though it can happen over short distances; van Dijk et al., 2024), you would not expect the abundance to be as high as observed *in-situ* at sites like in the Baltic Sea because of dilution and degradation of the cable bacteria, also during long-term burial.

Furthermore, Soledad basin is approximately 85 km long and 35 km wide and has a flat bottom (Silverberg et al., 2004). Our samples were taken far away from the slopes. There are no rivers in the vicinity of the basin and there is no evidence for major lateral input of sediment material to this basin, also not from prior work. Silverberg et al. (2004), for example, attributed variations in input of lithogenic material in sediment traps in Soledad basin to variations in aeolian dust from the continent – other inputs were considered negligible. Van Geen (2003) and others have shown that the sediment records from Soledad basin are highly suitable for detailed palaeoceanographic reconstructions. They also highlighted the similarity of many sediment records from different locations in Soledad basin, indicating lack of sediment heterogeneity, again confirming a limited role for lateral sediment input. To address the reviewer's point, we have expanded the site description, and we mention in the text that lateral transfer of cable bacteria is unlikely to explain our results.

Revised text:

Materials and methods:

L106-110: "Soledad basin is approximately 85 km long and 35 km wide and has a flat bottom. Sediments from different parts of the basin were previously found to be similar, indicating limited sediment heterogeneity within the basin (van Geen et al., 2003). There is little detrital sediment input (Chong et al., 2012). Sediment deposited in the basin is dominated by marine

snow, containing, among others, fecal pellets, pteropod shells and foraminifera from the overlying water column (Silverberg et al., 2004).

Discussion:

L432-437. "Importantly, lateral transfer of cable bacteria with sediment particles from shallower regions – which could theoretically contain cable bacteria - cannot explain the observed abundance of cable bacteria in these basins given (1) the low rate of deposition of detrital material in these basins (e.g. van Geen et al., 2003; Table 1), which, for Soledad basin where cable bacteria abundances are highest, is mostly aeolian (Silverberg et al., 2004) and (2) the unlikelihood of long-term survival of such a relatively high number of cable bacteria during long-distance transport and subsequent burial (van Dijk et al., 2024).

Section 4.2: The discussion of the sources of $H_2S$ and its temporal dynamics is stimulating. However, it remains hypothetical and suffers from a lack of (temporal) data, in my opinion. Have you tested correlations between the various parameters (i.e., $C_{org}$, total S, Fe sulphides, etc.) to perhaps highlight this dynamic and support a periodic (seasonal?) increase in sulphate reduction?

Reply: The temporal dynamics in organic matter input in the basins along the Californian and Mexican margin is firmly grounded in prior work for this region. This was noted in the original manuscript with appropriate references (For example: "Seasonal and interannual variability of primary productivity and, hence, of organic matter supply to the sediment is a common feature of the basins along the Californian and Mexican margin (Thunell, 1998; Silverberg et al., 2004; Collins et al., 2011)." And "Increased coastal upwelling in the region in winter and spring not only enhances primary productivity but also can lead to periodic, sometimes even seasonal inflows of oxygen into the basins, as reported for the Santa Barbara basin (e.g. Bograd et al., 2002; Peng et al., 2024)". We have expanded the information on the oceanographic setting in both the methods and the discussion section.

Non-steady state features are evident from the $C_{org}$ and total S profiles, as noted in the original manuscript. We have expanded the corresponding section. Unfortunately, further correlation analysis is not expected to provide useful information when applied to non-steady state diagenesis at sites with a low rate of sedimentation when there is temporal variability – as is the case here. This is because of diagenetic overprinting. As noted in the manuscript, we do see a link between average Fe/S ratios and cable bacteria densities, indicating that, in basins where there is a higher supply of S relative to Fe, we see more cable bacteria. This supports a role for periodic enhanced sulfate reduction. We expanded this section to clarify this.

Revised text:

Materials and methods:

L101-117: Sediment trap studies have shown that the input of organic matter to the basin varies both seasonally and interannually (Silverberg et al., 2004). Variations in coastal upwelling, which is a general feature of the region, are known to occur on seasonal, interannual and decadal time scales Generally, offshore Ekman transport and associated coastal upwelling are strongest

in the winter and spring, with upwelling becoming less strong in summer and fall, but still supporting a high primary productivity (Tems and Tappa, 2024). Paleoceanographic studies have confirmed variability in productivity in this setting (van Geen et al., 2003; Tems and Tappa, 2024). Given the relatively shallow water depth at the Soledad site (and at San Blas), changes in productivity will impact the input of organic matter to the sediment.

Discussion:

L461-465: "Such $C_{org}$ trends deviate from the typical downward decay profiles observed in sediments with a constant input of organic matter that are undergoing steady-state diagenesis (Burdige, 2006). The spikes in total S suggest variations in the in-situ production of $H_2S$ and its conversion to Fe sulfides, in line with non-steady state diagenesis (Burdige, 2006). Given the variability in productivity known for the region, temporal variability in the input of organic matter is the most likely explanation (Figure 4)."

Lines 434-447: in my opinion, this paragraph lacks any link with the biogeochemical data to explain the diversity observed.

Reply: In this section of the text, we provide details on the metabolic pathways of micro-organisms in the sediments at our sites that directly link with the data on, for example, the presence or absence of oxygen and ammonium since these are substrates for the microbes that we are discussing. We have expanded the text to clarify this.

Revised text: L506-508. "Distinct patterns emerged in the microbial community data, from which some biogeochemical inferences can be made, since the metabolic pathways imply the presence or absence of suitable substrates. Below, we detail how these inferences match with what we expect from the porewater and solid phase profiles (Figure 3 and 4)."

Line 441: what electron acceptor do the Bacteroidota use?

Reply: Bacteroidota defy simple characterization in terms of their electron acceptor use, with members capable of aerobic respiration, various anaerobic respiration pathways, as well as fermentation. We have added that they have a broad metabolic capability in the text.

Revised text: L 514-515: "These included Bacteroidota, which have a broad metabolic capability (aerobic and anaerobic respiration, fermentation) and are considered the primary degraders …".

 Line 453: it's a shame that we don't have this data, as it would have helped to underpin the discussion on interspecific competition.

Reply: We agree. This should be a target for further research.

Line 461: any suggestions on these factors (bioturbation or others generating sediment heterogeneity, Fe curtain, etc.)?

Reply: We have modified the text to clarify that with "different factors" we were referring to factors other than oxygen availability, i.e. we were referring to the preceding sentence in which we were discussing published work for the Santa Barbara basin. We have also added a sentence indicating that bioturbation is expected to hinder both cable bacteria and *Thioploca*. It is not clear to us what "Fe curtain" refers to in this context. The sediment heterogeneity in Soledad Basin is not high, see reply to an earlier comment above.

Revised text: L536-538:"…may be driven by other factors than the availability of oxygen. The lack of bioturbation, a common feature of sediments in ODZs (Levin, 2003) is beneficial to the establishment of both cable bacteria and *Thioploca* (Schulz and Jørgensen, 2001; Hermans et al., 2019)."

Lines 475-476: out of curiosity, do you have any idea what these benefits might be? The same for the selective pressures mentioned line 489. The section 4.4 is very interesting and frustrating: we want to know more!

Reply: Based on the line numbers, we infer that this statement refers to the co-existence of multiple species (Line 475), and the phenomenon of gigantism (paragraph starting Line 477). Regarding co-existence of multiple species, we agree this is intriguing and worthy of further investigation. Regarding the occurrence of gigantism in cable bacteria, we are uncertain if there are selective pressures driven by top-down forces (e.g., predation) or bottom-up forces (e.g. resource acquisition). Indeed, we would also like to know more and with the current text we are providing direction for future studies.

Section 4.5: this section could perhaps be further summarised and incorporated into the second paragraph of the conclusion.

Reply: While we appreciate the reviewer's comment, we prefer to keep this as an outlook section since we are introducing additional points of discussion and additional references and suggestions for further research that are not appropriate in a conclusion section.

Figures:

Figure 1: the blue colour contrast is poor. Would it be possible to improve it like in Bruggmann et al., 2023?

Reply: For reference and easy comparison, we repeat the Bruggmann et al. 2023 figure (top) and our figure (bottom) below. While we appreciate the reviewer's comment, we do not see the benefit of replacing our figure with that of Bruggmann et al. (2023).The figure would not provide additional insight in the bathymetry relevant to the relatively shallow sites of our study and in fact would lead to less information since the water depth differences between the stations are less well visible and the figure lacks a legend. We also note that our color scheme is the ODV standard and cannot be changed. Detailed information on the water depths is given in Table 1. We have added this information in the figure caption.

[Figure]

[Figure]

Revised text: L103: "…The water depths at the five stations are given in Table 1."

Figure 2: you write that triplicates were achieved for pH and H$_2$S μprofiles. Is the absence of standard deviation on these profiles justified by their high reproducibility? I suppose so for H$_2$S (because there isn't any), but what about pH?

Reply: Thank you for this comment. We indeed report only the first profile and have removed the mention of triplicates for pH and H$_2$S from the manuscript. See reply to earlier comment and modified text.

Figures 2, 3 and 4: please, put the unit of the vertical axis in cm.

Reply: We have changed "cmbsf", which refers to "cm below seafloor", by "cm" as suggested.

References: please see below and in the original manuscript

Daviray, M., Geslin, E., Risgaard-Petersen, N., Scholz, V. V., Fouet, M., and Metzger, E.: Potential impacts of cable bacteria activity on hard-shelled benthic foraminifera: implications for their interpretation as bioindicators or paleoproxies, Biogeosciences, 21, 911–928, https://doi.org/10.5194/bg-21-911-2024, 2024.

Scholz VV, Muller H, Koren K, Nielsen LP, Meckenstock RU. 2019. The rhizosphere of aquatic plants is a habitat for cable bacteria. FEMS Microbiology Ecology 95: fiz062.

Scholz, V.V., Martin, B.C., Meyer, R., Schramm, A., Fraser, M.W., Nielsen, L.P., Kendrick, G.A., Risgaard-Petersen, N., Burdorf, L.D.W. and Marshall, I.P.G. (2021), Cable bacteria at oxygen-releasing roots of aquatic plants: a widespread and diverse plant–microbe association. New Phytol, 232: 2138-2151. https://doi.org/10.1111/nph.17415

Tems, C.E.; Tappa, E. Regional Fluctuations in the Eastern Tropical North Pacific Oxygen Minimum Zone during the Late Holocene. Oceans 2024, 5, 352–367.

van Dijk, J.R., Geelhoed, J.S., Ley, P., Hidalgo-Martinez, S., Portillo-Estrada, M., Verbruggen, E. et al. Cable bacteria colonise new sediment environments through water column dispersal. Environmental Microbiology, 26(10), e16694, 2024

**Response to reviewer #2**

**Review of Slomp et al. A niche for diverse cable bacteria in continental margin sediments overlain by oxygen-deficient waters.**

The manuscript of Slomp et al. describes a study where activity and abundance of Cabel bacteria were addressed in sediments from 5 hypoxic basins along the coast of Mexico and California, together with a multitude of geochemical features ($O_2$, pH, Fe, Mn, Nox, Al ) and microbial populations. In general cable bacteria abundance were low and activity were below detection limit. Phylogenetic analysis revealed that the Cable bacteria belonged to the Candidatus Electrotrix lineage and included specimens affiliated with *Electrotrix gigas*. In addition several sequences were affiliated with a sister clade to Electrotrix and it is suggested that these represent a novel genius.

In general, the manuscript adds novel information about the biogeography and diversity of cable bacteria in marine environments and has therefore merits justifying publication.

Reply: We thank the reviewer for the positive words

I recommend however that the authors put somewhat more effort in discussing the vast amount of geochemical data and provide the motivation behind the analysis these features. There are severall fractions of solidphase and dissolved compounds that is not (to my knowledge )really

linked to cable bacteria (e.g. Al, Mn, NH4 ): Why is this relevant in the given context?; What was the hypothesis?

Reply: Indeed, we present a wealth of geochemical data. These data are relevant to understanding the factors controlling the electron acceptor and donor availability for cable bacteria in this setting (i.e. the availability of oxygen, nitrate and sulfide). Regarding the elements mentioned: the dynamics of Fe and Mn are closely linked to those of sulfide and oxygen. Aluminum is used to normalize Mn and Fe to assess for dilution of reactive Fe and Mn by detrital components. Checking for such dilution is standard in geochemistry. Fe/Al ratios are valuable as a redox proxy. Porewater $NH_4$ is an excellent indicator of the rate of degradation of organic matter – more so than the organic carbon records, which just reflect the proportion of the organic matter that has not degraded yet. We also note that geochemical records can carry signatures of cable bacteria activity (Risgaard-Petersen et al., 2012), i.e. we know that cable bacteria can impact Fe, Mn and S dynamics. We have added text in the methods and discussion to further clarify the points above. Line numbers refer to the track changes version of the manuscript.

Revised text: L215-226: "A range of parameters relevant to Fe, Mn and S cycling were determined. These parameters provide additional constraints on the bottom water redox conditions at our sites and the degree of reaction with sulfide. Aluminum was used to normalize Fe and Mn to assess for dilution of reactive Fe and Mn by detrital components. Average ratios of Fe/Al (wt%/wt%) were calculated per site to assess the potential enrichment of the sediment with Fe relative to a detrital background, which ideally is defined locally but often lies close to 0.55 (Raiswell et al., 2018). Taking the sum of all Fe fractions from the Fe speciation as a measure of highly reactive Fe (FeHR), we calculated the average fraction of the total Fe per site that is highly reactive (FeHR/Fetotal) and its pyritized fraction (FePYR/FeHR) (Poulton and Canfield, 2011; Raiswell et al., 2018). These fractions are common indicators for bottom water redox conditions, with higher values pointing towards increased anoxia/euxinia (Raiswell et al., 2018). Ratios of Fe/S were determined as a measure of the degree of sulfidization of the Fe, with lower Fe/S ratios pointing to higher degrees of sulfidization (e.g. Kraal et al., 2017)."

L481-484: "The relatively low degree of pyritization of the Fe (FePYR/FeHR <0.3), at the low end of the range for anoxic and ferruginous environments (0.22-0.7) is consistent with such an environment (Poulton and Canfield, 2011; Raiswell et al., 2018). The Fe/Al ratios are variable and do not provide additional constraints on the bottom water redox conditions in this setting."

L470: "The porewater profiles of $NH_4^+$, which can be used as an indicator of anaerobic organic matter degradation, are in line with this trend (Figure 3). Soledad, in particular, stands out as a site with high rates of anaerobic metabolism and high numbers of cable bacteria."

In addition the motivation behind the analysis of the general microbial community should be stated more clear. What is the relevance of this analysis for the target features: cable bacteria?

Thank you for your question and the opportunity to elaborate. By analyzing cable bacteria together with the broader microbial community, we gain a better understanding of the factors governing cable bacteria distribution and activity, providing a more comprehensive understanding of their biogeochemical role. We report on the microbial community composition in our results and

supplemental material and discuss the composition in some detail (section 4.3), because it provides insight into the biogeochemical functioning of the study sites, which sets up the context for interpreting the niche of Electrothrix. The microbial community composition, characterized by highly structured beta diversity patterns (Fig. 6), and displayed with taxonomic resolution in the supplemental material, enables inferences about the metabolic processes occurring across our sampling sites. These biogeochemical conditions effectively define the niche parameters for the cable bacteria. By combining the microbial community data with the comprehensive biogeochemical measurements, we can formulate testable hypotheses regarding the physiological capabilities and adaptations of cable bacteria in these poorly understood environments. We have added text in the discussion to make this more explicit.

Revised text L495-497: "By analyzing cable bacteria together with the broader microbial community, we gain a better understanding of the factors governing cable bacteria distribution and activity. This provides a more comprehensive understanding of the biogeochemical role of cable bacteria and their niche parameters."

L505-508: "Distinct patterns emerged in the microbial community data, from which some biogeochemical inferences can be made, since the metabolic pathways imply the presence or absence of suitable substrates. Below, we detail how these inferences match with what we expect from the porewater and solid phase profiles (Figure 3 and 4)."

L520-522: "These findings reinforce the idea (Section 2.1) that the bottom-water oxygen concentrations during our sampling campaign are representative of persistent differences between sites and hence contribute to our understanding of the occurrence of a niche for cable bacteria in this environment."

In my view these data should be more integrated in the overall framework of the study:

Would it e.g. be possible to apply e.g. network analysis to identify specific cablebacteria – microbe associations?

Reply: Unfortunately, this dataset is not well-suited to network analysis due to the sample number and sparsity of the cable bacteria. Liu et al. 2021 were able to successfully examine the networks of cable bacteria (in their system), but this was from a laboratory experiment with a total of 60 samples. In our case, this approach would not have sufficient replication to be robust.

Minor:

L 32: remove the Nielsen et al . 2010 reference.  This paper says nothing about cable bacteria. The authors suggest nanowires or conductive minerals as mediators of the electric currents that runs through the sediment (exactly like in the Revil et al. (2010) paper!) The discovery of cable bacteria  was published in 2012. i.e. two years after the Nielsen et al. paper.

Reply: We have made the change as suggested.

L 34: Remove the reference Nielsen and Risgaard-Petersen 2015 paper: This is a review paper that amongst other review the Risgaard-Petersen et al. 2012 paper on how cable bacteria influence the biogeochemistry.

Reply: We have made the change as suggested

L 49. The Damgaard et al 2014 paper describes the construction and application of the silver silver chloride electrode used for electric potential measurements. The focus in this paper is not relationships between cable bacteria activity and the electric potential. Such is more thoroughly described in Risgaard-Petersen et al. (2012) and in Risgaard-Petersen et al. (2014). Since both papers precedes the Damgaard et al. paper I suggest that that the authors include those here instead of the Damgaard et al. paper.

Reply: We have made the change as suggested.

L 133. Please spell out "EP": Electric potential.

Reply: EP is spelled out at its first occurrence in line 115 (original manuscript).

L150: What sediment volume was collected for FISH: 0.5 ml?? Please specify.

Reply: We added that this was a volume of 0.5 mL in line 167.

L 195. The method description for quantitative FISH analysis is a bit odd. What is the rational for not using direct methods like those described in e.g. Schauer et al. (2014)?. Why use Nycodens extractions? There is to my experience a great risk of loosing cells with this method. Do the authors have any data on the cell recovery efficacy ?

Reply: The density gradient methods described have been used in several previous field and laboratory studies investigating cable bacteria (e.g, Malkin et al. 2022, Liau et al. 2022). Trojan et al. 2016, is the first study we are aware of that used this method with Nycodenz, based on the procedure published by Kallmeyer et al. (2008). Trojan et al. 2016 specifically used this method on samples they obtained from Aarhus Bay. In our revision, we added that reference to the corresponding part of the text. Our rationale was based on our laboratory experience: we have found that we generally achieve the same or higher counts with this method, over the less labor-intensive method of simply adding sediment to a slide. This is effective presumably because it removes particles that impede the view of cable bacteria. We note that a critical step is that we collect the supernatant on a filter after each wash, minimizing cell loss between washing steps. Although that information is stated in Malkin et al. 2022, and Kallmeyer et al. 2008, we will repeat that information in this manuscript for clarity.

Revised text: L229-236: "Cable bacteria were enumerated through fluorescence in-situ hybridization (FISH) as described in detail by Malkin et al. (2022), adapted from Trojan et al. (2016). Briefly, cells were detached from sediment particles using acetate buffer to dissolve carbonates, followed by repeated washes with NaCl solution (collecting the washing and supernatant each time). The samples were then resuspended in a detergent (Tween 80) with 10%

methanol solution and vortexed for 60 minutes. Density centrifugation with 50% (w/v) Nycodenz solution was subsequently used to separate detached microbial cells from sediment particles (Kallmeyer et al., 2008). Cells from both the washing supernatant and the aqueous layer from centrifugation were captured on a cellulose acetate filter (0.2 μm), which was kept frozen (-20°C) until further analysis. Filters were sectioned by razor blade and sections were subjected to FISH staining."

It apereas that Sybr Green I staining were used for Cable bacteria quantification : Why this step when you have FISH stained filaments already?

Reply: Thank you for this comment. We indeed did not additionally use Sybr Green. All samples examined for this study were FISH-stained. This sentence has been removed from the text.

References in replies above: please see the original manuscript.

References

Revil, A., Mendonca, C.A., Atekwana, E.A., Kulessa, B., Hubbard, S.S., Bohlen, K.J., 2010. Understanding biogeobatteries: Where geophysics meets microbiology. J. Geophys. Res. (Biogeosci.) 115, G00G02.

Risgaard-Petersen, N., Damgaard, L.R., Revil, A., Nielsen, L.P., 2014. Mapping electron sources and sinks in a marine biogeobattery. J. Geophys. Res. (Biogeosci.) 119 1475–1486.

Risgaard-Petersen , N., Revil, A., Meister, P., Nielsen, L.P., 2012. Sulfur, iron-, and calcium cycling  associated with natural electric currents running through marine sediment. Geochimica Et Cosmochimica Acta 92, 1-13.

Schauer, R., Risgaard-Petersen, N., Kjeldsen, K.U., Bjerg, J.J.T., Jørgensen, B.B., Schramm, A., Nielsen, L.P., 2014. Succession of cable bacteria and electric currents in marine sediment. Isme Journal 8, 1314-1322.